# Interactions of Chemically Synthesized Ferrihydrite Nanoparticles with Human Serum Transferrin: Insights from Fluorescence Spectroscopic Studies

**DOI:** 10.3390/ijms22137034

**Published:** 2021-06-29

**Authors:** Claudia G. Chilom, Nicoleta Sandu, Sorina Iftimie, Maria Bălăşoiu, Andrey Rogachev, Oleg Orelovich, Sergey Stolyar

**Affiliations:** 1Department of Electricity, Solid Physics and Biophysics, Faculty of Physics, University of Bucharest, Str Atomistilor 405, CP MG 11, RO-077125 Măgurele, Romania; s.nicoleta59@yahoo.ro (N.S.); sorina.iftimie@fizica.unibuc.ro (S.I.); 2Joint Institute for Nuclear Research, Joliot-Curie No. 6, 141980 Dubna, Russia; masha.balasoiu@gmail.com (M.B.); andrey.v.rogachev@gmail.com (A.R.); orel@jinr.ru (O.O.); 3Horia Hulubei National Institute for R&D in Physics and Nuclear Engineering, RO-077125 Măgurele, Romania; 4Moscow Institute of Physics and Technology, Institutskiy Per. No. 9, 141701 Dolgoprudniy, Russia; 5Krasnoyarsk Science Center of the Siberian, Branch of the Russian Academy of Sciences, Akademgorodok St. No. 50, 660036 Krasnoyarsk, Russia; rauf@iph.krans.ru

**Keywords:** ferrihydrite nanoparticles, human serum transferrin, binding mechanism, driving forces, molecular docking

## Abstract

Human serum transferrin (HST) is a glycoprotein involved in iron transport that may be a candidate for functionalized nanoparticles to bind and target cancer cells. In this study, the effects of the simple and doped with cobalt (Co) and copper (Cu) ferrihydrite nanoparticles (Fh-NPs, Cu-Fh-NPs, and Co-Fh-NPs) were studied by spectroscopic and molecular approaches. Fluorescence spectroscopy revealed a static quenching mechanism for all three types of Fh-NPs. All Fh-NPs interacted with HST with low affinity, and the binding was driven by hydrogen bonding and van der Waals forces for simple Fh-NPs and by hydrophobic interactions for Cu-Fh-NPs and Co-Fh-NPs binding, respectively. Of all samples, simple Fh-NPs bound the most to the HST binding site. Fluorescence resonance energy transfer (FRET) allowed the efficient determination of the energy transfer between HST and NPs and the distance at which the transfer takes place and confirmed the mechanism of quenching. The denaturation of the HST is an endothermic process, both in the case of *apo* HST and HST in the presence of the three types of Fh-NPs. Molecular docking studies revealed that Fh binds with a low affinity to HST (*K_a_* = 9.17 × 10^3^ M^−1^) in accord with the fluorescence results, where the interaction between simple Fh-NPs and HST was described by a binding constant of 9.54 × 10^3^ M^−1^.

## 1. Introduction

Human serum transferrin (HST), a member of the sub-family of serum transferrins, is an 80 kDa bilobal iron transport glycoprotein, with 697 amino acid residues (of which three are tryptophan (Trp) and 26 are tyrosine (Tyr)) [1,2]. HST structure is organized in two similar but not identical lobes, the N-terminal and C-terminal lobes [3]. The HST concentration can be measured indirectly from the maximum iron-binding capacity of plasma. The reference value for an adult is in the range 250–400 μg/dL and 30% is saturated with iron [4,5]. In addition to transporting iron, HST binds and transports a wide variety of molecules such as flavonoids [6,7] and vitamins [8]. HST may be a candidate for functionalized NPs to target cancer cells as they have an increased demand for iron and stimulate TFRC expression [9].

Ferrihydrite (Fh), (Fe^3+^)_2_O_3_•0.5H_2_O, an iron oxide, can be synthesized [10,11], as NPs with structural features and properties that can be explored in biophysical and medical applications. One of the directions of research in biophysics is the interaction with biological structures such as proteins [12,13], liposomes and natural membranes [14], and cells [15]. In the medical field, the magnetic properties of the iron nanoparticles, including ferrihydrite nanoparticles (Fh-NPs), have been exploited in the field of the immunoassay [16], magnetic resonance imaging (MRI) [17], and magnetic particle imaging (MPI) for noninvasive imaging of inflammation processes [18], drug delivery [19], and hyperthermia [20]. The doping of nanoparticles can lead to an extension of their field of application by improving the physicochemical characteristics [21] such as the modification of their magnetic properties. The doping of NPs with copper improves the catalytic properties of NPs as do noble metals, but at lower costs. In the case of ferrihydrite nanoparticles, it was found that the doping with cobalt leads to an increase in the anisotropy constant of the nanoparticles and to the formation of surface rotational anisotropy [11]. An increase in the magnetic susceptibility of ferrihydrite nanoparticles by doping [10] opens the way to determine optimal sizes of nanoparticles for medical applications.

The aim of this study is to characterize the binding mechanism of simple and doped Fh-NPs with HST protein. The goal is to achieve information regarding the nature of the binding mechanism, the forces that drive the interaction, but also the effect of NPs on the thermal stability of the HST protein is investigated. In this sense, a spectroscopic approach was chosen, supplemented with molecular docking. One of the criteria on the basis of which ferrihydrite nanoparticles were chosen was their biocompatibility. This mineral is the core of the ferritin protein complex, main carrier of iron in higher living organisms [22], and is characterized by a high reactivity and high ion adsorption capacity [23]. Also, ferrihydrite nanoparticles have a positive effect on the activation of phagocytosis in the liver, stimulate the proliferation and intracellular regeneration of hepatocytes [24], and are used in the environmental remediation of water and soils [25].

## 2. Materials and Methods

Human serum transferrin was purchased from Merck Company. Protein stock concentration was 5 × 10^−4^ M prepared in 100 mM HEPES buffer at pH 7.35.

Ferrihydrite nanoparticles: Synthetic ferrihydrite particles, simple and doped with cobalt (Co) and copper (Cu), were prepared as described earlier [11]. In brief, chemically synthetic ferrihydrite nanoparticles were prepared as following: a 1 M alkaline solution of NaOH was added to a 0.2 M solution of iron nitrate Fe(NO_3_)_3_·9H_2_O, at room temperature, under constant stirring, until a value of pH equal to 7.0 was achieved. The precipitate was collected on a filter, washed, and dried at room temperature. Ferrihydrite particles doped with Cu (Cu-Fh-NPs) and Co (Cu-Fh-NPs) were prepared at room temperature, by the slowly adding of the alkaline solution of NaOH to iron nitrate, and of a solution of Co or Cu salts that was added with continuous stirring to achieve a value of pH equal to 7.0. The pH of the solution was monitored using a pH-meter. The precipitate of doped NPs was collected on the filter and was repeatedly washed and dried at room temperature.

The obtained particles have been characterized earlier by means of atomic force microscopy (AFM), transmission electron microscopy (TEM), scanning electron microscopy (SEM), and small-angle neutron scattering methods (SANS) [12,13]. The results obtained in previous studies using Scanning Electron Microscopy (SEM) showed that samples investigation present normal size distribution particles with 6.4 nm mean value and 0.94 standard deviation for the Fh-NPs sample, 10.5 nm and 1.65 in the case of Co-Fh-NPs particles and 9.5 nm and 1.47 standard deviation for Cu-Fh-NPs [13]. Structural characterization of ferrihydrite samples by Transmission Electron Microscopy (TEM) also showed values below 10 nm of simple and doped Fh-NPs [11]. Atomic Force Microscopy (AFM) showed a roughly spherical shape of ferrihydrite nanoparticles with average particle height for simple nanoparticles 6.11 ± 4.51 nm with nanoparticles varying between 2 nm and 30 nm, 6.22 ± 3.27 nm for Co-Fh-NPs with heights up to 20 nm and 7 ± 3.47 nm for Cu-Fh-NPs with a narrower distribution between 3 and 16 nm [14]. From the small-angle neutron scattering (SANS) measurements ellipsoidal agglomerates were determined with the smallest dimensions of ~9 nm (Fh-NPS), ~14 nm (Cu-Fh-NPS), and ~13.5 nm (Co-Fh- NPS) [12].

Fluorescence spectra were recorded using a Perkin Elmer LS55 spectrofluorometer, equipped with a thermostatically controlled holder. The excitation was set at 295 nm (in order to avoid Tyr excitation), and the emission was collected between 310 and 400 nm, with 500 nm/min speed scan. The slits width for both excitation and emission wavelengths was set at 10 nm. All spectra were corrected in order to eliminate the inner filter effect. Fluorescence intensity changes of HST and HST-simple or doped Fh-NPs complexes were monitored against heat denaturation at the excitation wavelength of 295 nm. HST and HST-Fh-NPs complex, at a concentration of 3 μM HST and 10 μM NPs, were dissolved in 100 mM HEPES buffer (pH 7.35).

Fluorescence resonance energy transfer (FRET) was tracked by overlapping donor emission spectrum (HST) with acceptor absorption spectrum (Fh-NPs). The distance between the nanoparticles and HST was calculated according to Equation (1):(1)$$E=1-\frac{F}{F_{0}}=\frac{R_{0}^{6}}{R_{0}^{6}+r^{6}}$$
where *E* is the efficiency of energy transfer, *F*_0_ and *F* are the fluorescence intensity of HST in the absence, respectively, in the presence of NPs, *r* is the distance from the bound NPs on HST binding site, and *R*_0_ (in Ǻ) is the Förster critical distance at which 50% of the excitation energy is transferred from the donor protein to the NPs. This distance can be determined using Equation (2):(2)$$R_{0}=9.78\times 10^{3}{\left[\left(k^{2}n^{-4}Q_{D}J\left(\lambda \right)\right)\right]}^{1/6}$$
where *k* expresses the relative orientation of the transition dipoles of the protein and NPs, *n* is the refractive index of medium, *Q_D_* is the quantum yield of HST fluorescence, in the absence of NPs, and *J* is the overlap integral of the emission spectrum of the protein with the absorption spectrum of the NPs, calculated by Equation (3):(3)$$J=\frac{\int^{\;}F_{D}\left(\lambda \right)\varepsilon_{A}\left(\lambda \right)\lambda^{4}d\lambda }{\int^{\;}F_{D}\left(\lambda \right)d\lambda }$$
where *F_D_* (*λ*) is the corrected fluorescence intensity of HST at wavelength *λ* with the total intensity (area under the curve) normalized to unity, and *ε_A_* (M^−1^ cm^−1^) is the molar absorption coefficient of the acceptor at the wavelength *λ*.

Molecular docking: The crystal structure of human serum transferrin (1D3K) was retrieved from Protein Data Bank [26,27,28]. The structure of the ferrihydrite (the structure of six-line ferrihydrite) was retrieved from Crystallography Open Database [29,30]. Ferrihydrite (Fh), (Fe^3+^)_2_O_3_•0.5H_2_O, an iron oxide, can be synthesized [10,11], as NPs with structural features and properties that can be explored in biophysical and medical applications. The docking calculations, determining the binding affinity and the optimal orientation of the ligand at the protein site, were possible using *PyRx* [31] with the *Auto Dock Vina* [32] algorithm. *UCSF Chimera,* developed by the Resource for Biocomputing, Visualization, and Informatics from University of California, San Francisco, with support from NIH P41-GM103311 [33] was also used in the docking process. During the simulation, the size of the grid box along x, y, and z directions were all set at 25 Å. The grid box center was set at (47.6, 45.6, 43.8) and the exhaustiveness equaling to 100 were carried out to obtain the possible binding conformations.

## 3. Results and Discussions

Based on the property of transferrin to bind iron, we consider Fh-NPs to be a good candidate that can be bound to HST to be transported through plasma to cells. The study of the binding mechanism can be done by monitoring the changing in the fluorescence of HST when simple and doped Fh-NPs were added.

### 3.1. Fluorescence Quenching of Human Serum Transferrin by Fh-NPs

The mechanism of binding of simple and doped Fh-NPs to HST was investigated by fluorescence, based on the fluorescent properties of the tryptophan (Trp) residues. By increasing the concentration of simple and doped nanoparticles (0–15) µM, the fluorescence of HST (3 µM) decreased (Figure 1A). The same quenching behavior was obtained for the titration of Fh-NPs doped with Cu and Co (Appendix A) in HST solution. This quenching of the HST fluorescence is due to the binding of Fh-NPs to HST site. The study of the quenching mechanism of HST by Fh-NPs was performed based on the Stern-Volmer equation (Equation (4))
(4)$$F_{0}/F=1+K_{q}\tau_{0}\left[Q\right]=1+K_{SV}\left[Q\right]$$
where $F_{0}$ and F are the fluorescence intensities of HST in the absence and in the presence of the quencher (NPs, in this case), respectively; [Q] is the concentration of the quencher; $K_{q}\;$ is the bimolecular quenching rate constant; $\tau_{0}$ is the average lifetime of the protein in the absence of the quencher, and $K_{SV}$ is the quenching constant (Stern-Volmer constant). The bimolecular quenching rate constant ($K_{q}$) was calculated as the ratio between the quenching constant (Stern-Volmer constant) and the average lifetime ($\tau_{0}$) of the protein in the absence of the quencher (3.34 ns) [34].

The Stern Volmer representation at 25 °C (Figure 1B) and also at 35 °C (Appendix A) is linear for Fh-NPs-HST and non-linear for Cu-Fh-NPs-HST and Co-Fh-NPs-HST, respectively. The results for Fh-NPs-HST suggest a static quenching mechanism because the values of the Stern-Volmer constants are higher at 25 °C than at 35 °C. In the case of the other two complexes, Cu-Fh-NPs-HST and Co-Fh-NPs-HST the Stern-Volmer plot shows downward curvatures [35]. Such a representation indicates that at low concentrations of doped NPs the intensity of the emitted light is low and the extinction of HST fluorescence can be observed. As the concentration of NPs doped with Cu and Co increases, the intensity of the emitted light increases slowly. This nonlinear dependence can be explained by the existence in solution of two species that emit differently, depending on their exposure to the solvent [36]: one free, uncomplex species and another in the ground state complex. The F_0_/F curve can be fitted by a single exponential, which indicates a single type of HST fluorescence quenching by doped NPs. Moreover, the values of the bimolecular constants (Table 1) are higher than the value for the diffusion limit in aqueous solutions, 1 × 10^10^ M^−1^ s^−1^ [37], which is an indication that the main binding mechanism is static. The downward curvature present in the case of doped NPs may be a consequence of HST complexation with Cu or Co detaching ions from NPs. In this way, the complexation of the protein with ions produces conformational changes in the structure of HST (in which fluorophores are in inaccessible positions), whose fluorescence can no longer be quenching by NPs. The apparent quenching constant, corresponding to the static component of the interaction was determined for the first experimental data that can be fitted with a straight line. The results are presented in Table 1.

### 3.2. Binding Mechanism and Driving Forces of Fh-NPs and Human Serum Transferrin

The binding mechanism of the three types of Fh-NPs to HST was characterized based on the Scatchard equation (Equation (5)) at 25 °C (Figure 1C) and 35 °C (Appendix A).
(5)$$\log(\frac{F_{0}}{F}-1)=\log K_{b}+n\log[Q]$$
where $F_{0}$ and F are the fluorescence intensities of HST in the absence and presence of the NPs, respectively; $K_{b}$ is the binding constant; [Q]  is the NPs concentration, and n the stoichiometry of the binding. Considering *n* = 1, as Fh binds to serum proteins at one site [13,15], the binding affinity for HST was found to be 9.54 × 10^3^ M^−1^ for Fh-NPs, 5.10 × 10^3^ M^−1^ for Cu-Fh-NPs, and 1.30 × 10^3^ M^−1^ for Co-Fh-NPs. Thus, at 25 °C, the binding of all types of Fh-NPs to HST takes place with low affinity. Similar data were determined at 35 °C, and the results were listed in Table 1.

The relationships between the thermodynamic parameters of the protein-ligand interaction are related to the forces that drive the binding. The determination of thermodynamic parameters (Equations (6) and (7)) allowed a more complex characterization of the interactions that take place between protein and NPs.
(6)$$\ln\frac{K_{b2}}{K_{b1}}=\frac{\Delta H}{R}\left(\frac{1}{T_{1}}-\frac{1}{T_{2}}\right)$$
(7)$$\Delta G=-RT\ln K_{b}=\Delta H-T\Delta S$$
where *K_b_* is the association constant; *T* is the absolute temperature; *R* is the gas constant (8.314 J K^−1^ mol^−1^).

The thermodynamic parameters are related with the driving forces of the interaction: (*i*) if ΔH > 0 and ΔS > 0, the main driving forces are hydrophobic interaction; (*ii*) if ΔH < 0 and ΔS < 0, then the interaction is driven by the hydrogen bonding and van der Waals forces; (*iii*) if ΔH < 0 and ΔS > 0, the interaction is driven by electrostatic forces [38,39]. Following the interpretation of the obtained results (Table 1; Figure 1D), it was found that all processes are spontaneous (Δ*G* < 0), and in the case of Fh-NPs the main forces of the binding are hydrogen bonding and van der Waals forces (ΔH < 0 and ΔS < 0). For the binding of Cu-Fh-NPs and Co-Fh-NPs to HST (Table 1), processes are mainly driven by hydrophobic interactions (Δ*H* > 0 and ΔS > 0).

To better understand the effect of the simple Fh-NPs and doped with Cu and Co on different Trp residues of HST, the second derivative of fluorescence spectra obtained by NPs titration to HST were analyzed (Figure 2A–C).

In the presence of simple and doped Fh-NPs, by increasing their concentrations (2, 5, and 10 μM), the minimum of the second derivative spectrum of HST was red shifted to higher wavelengths. This is an indication that Trp microenvironments are influenced by the interaction with each of Fh-NPs. The shifts in the maximum wavelength for Tyr and Trp in the presence of high NPs concentrations can be measured and the magnitude and direction of the peak shifts are influenced by the type of NPS, the local environment, and solvent accessibility of the Tyr and Trp. It was observed that with increasing affinity for binding of NPs to the HST site, the rightward displacement increased as follow: Fh-NP > Cu-Fh-NPs > Co-Fh-NPs (Figure 2D). The red shift of the spectra is related to the increase of the solvent polarity.

### 3.3. The Effect of Ferrihydrite Nanoparticles on the Thermal Stability of HST

In order to evaluate the role of simple and doped Fh-NPs in the thermal stability of HST, fluorescence of Trp was followed in the presence of a single concentration of each type of FhNPs (10 µM). The effect of the simple and doped Fh-NPs on HST thermal stability was investigated between 25 °C and 80 °C (Figure 3A), and the influence of the NPs on the HST denaturation process is illustrated in Figure 3B, where the maxima of HST fluorescence intensities are represented. Similar results were obtained when simple and doped Fh-NPs were added to HST solution (data not shown).

Considering that the process of thermal denaturation of HST assumes the model in two states, a native (*N*) and a denatured (*D*) states, the equilibrium of this process will be established according to the relation (8):(8)$$K_{eq}=\frac{\left[D\right]}{\left[N\right]}$$
where *N*-native (folded) and *D*-denatured (unfolded) are the two states of the protein structure.

Using the van’t Hoff equation (Equation (9)), ln*K_eq_* vs 1/*T* was represented (Figure 3C) and the values of melting temperature, *T_m_*, and Δ*H_unf_* and Δ*S_unf_* were obtained (Table 2).
(9)$$lnK_{eq}=-\frac{\Delta H}{RT}+\frac{\Delta S}{R}$$
where Δ*H* and ΔS are the variation of the enthalpy and of the entropy, respectively, *T* is the temperature and *R* is the gas constant.

The *T_m_* of the HST protein was about 54.10 °C in the absence of Fh-NPs, 52.10 °C in the presence of Fh-NPs, 52.63 °C in the presence of Co-Fh-NPs, and 54.14 °C in the presence of Cu-Fh-NPs. The obtained values of *T_m_* suggest that simple and doped Fh-NPs to HST cause small alterations in the protein structure. The thermodynamic parameters Δ*H* and Δ*S* are also listed in Table 2 for each denaturation process.

All types of Fh-NPs cause small alterations in the protein structure, which means that the structure of the HST protein remains stable in the presence of these NPs. This can be a starting point for using NPs in medical applications.

### 3.4. Fluorescence Resonance Energy Transfer from HST to the Ferrihidrite Nanoparticles

Simple Fh-NPs and doped with Cu and Co interact with HST via a static mechanism. In order to investigate whether an energy transfer between HST (3 µM) and Fh-NPs (simple and doped with Cu and Cu) is also possible, the fluorescence emission spectrum of HST (the donor protein) was overlapped with the absorption spectrum of each type of Fh-NPs (3 µM) (Figure 4).

For HST-Fh-NPs, the value of the overlap integral, J, was found to be 3.83 × 10^14^ M^−1^ cm^−1^ nm^4^. Considering k^2^ = 2/3, n = 1.336, and Q_D_ = 0.025 [34] in Equation (2), the distance R_0_ was calculated as 2.29 nm. From the Equation (1), the efficiency of energy transfer, E, was determined, E = 0.12, and also, the distance between HST and Fh-NPs was found to be 3.18 nm. This value, in agreement with the rule 0.5R0 < r < 1.5R0, confirms that simple Fh-NPs quenched the HST fluorescence via a static process [7]. The parameters were listed in Table 3, together with the parameters determined for Cu-Fh-NPs-HST and Co-Fh-NPs-HST.

In the case of the interaction between doped Fh-NPs and HST, the value of the overlap integral, *J*, was found to be 2.09 × 10^14^ M^−1^ cm^−1^ nm^4^ for Cu-Fh-NPs-HST and 1.82 × 10^14^ M^−1^ cm^−1^ nm^4^ for Co-Fh-NPs-HST. The distance *R*_0_ was found to be 2.07 nm for Cu-Fh-NPs-HST and 2.02 nm for Co-Fh-NPs-HST. Also, the efficiency of energy transfer, *E*, was determined as 0.08 for Cu-Fh-NPs-HST and 0.05 for Co-Fh-NPs-HST. Therefore, the distance between HST and Cu-Fh-NPs was found to be 3.10 nm and 3.23 nm for Co-Fh-NPs (Table 3). As the rule 0.5*R*_0_ < *r* < 1.5*R*_0_ is satisfied with the results obtained, we can admit that the interaction of these NPs with HST is due to a static process, confirming the results obtained in the fluorescence experiments. Also, from the obtained results we can say that the doping of the nanoparticles leads to the decrease of the efficiency (Table 3).

### 3.5. Molecular Docking between Ferrihydrite and HST

Molecular docking studies were performed by taking ferrihydrite (Fh) as a ligand, in order to predict the experimental binding mode and affinity of Fh within the binding site of HST. The docking simulation allowed viewing the optimal leads superimposed core cavity interaction between Fh and HST (Figure 5). Fh binds near the following residues in HST: Ala 191, Arg 124, Asn 213, Asp 292, Asp 297, Gln 206, Glu 212, Gly 190, Gly 187, Gly 290, His 14, His 207, His 249, Ile 210, Leu 293, Leu 182, Lys 291, Lys 296, Lys 193, Phe 186, Phe 192, Ser 189, Ser 208, Ser 298, Thr 209, Thr 181, Thr 120, Trp 8, Tyr 188, and Tyr 45. Therefore, the HST binding site for Fh contains the Trp 8 residue, whose modification of the microenvironment as a result of the interaction with Fh has been previously followed by fluorescence.

The binding affinity of Fh for HST structure of ∆G = 22.59 kJ/mol and the apparent association constant of K_a_ = 9.17 × 10^3^ M^−1^ were found. These results are in accord with the fluorescence investigations, for which the interaction of simple Fh-NPs with HST was described by a weak binding constant of 9.54 × 10^3^ M^−1^ and a ∆*G* = −22.71 kJ/mol at 25 °C.

## 4. Conclusions

In this study, we used fluorescence and molecular docking simulation to characterize the binding of the simple and doped ferrihydrite nanoparticles to human serum transferrin, HST.

The results obtained indicate that all three types of Fh-NPs bind to one site of HST by a static mechanism in the case of simple Fh-NPs, and by a combined mechanism, in the case of Cu-Fh-NPs and Co-Fh-NPs, respectively. The interaction takes place with small affinities and is driven by hydrogen bonding and van der Waals forces in the case of Fh-NPs, and by hydrophobic interactions in the case of Cu-Fh-NPs and Co-Fh-NPs binding. The distances between Fh-NPs, Cu-Fh-NPs, and Co-Fh-NPs obtained in FRET confirmed the static quenching mechanism. Molecular docking verified the results obtained in fluorescence.

The stability of HST structure is not influenced by simple and doped Fh-NPs, as the NPs caused only small alterations in the protein structure.

The results of this study may be relevant for the further development of ferrihydrite nanoparticles as diagnostic and therapeutic tools.

## Figures and Tables

**Figure 1 ijms-22-07034-f001:**
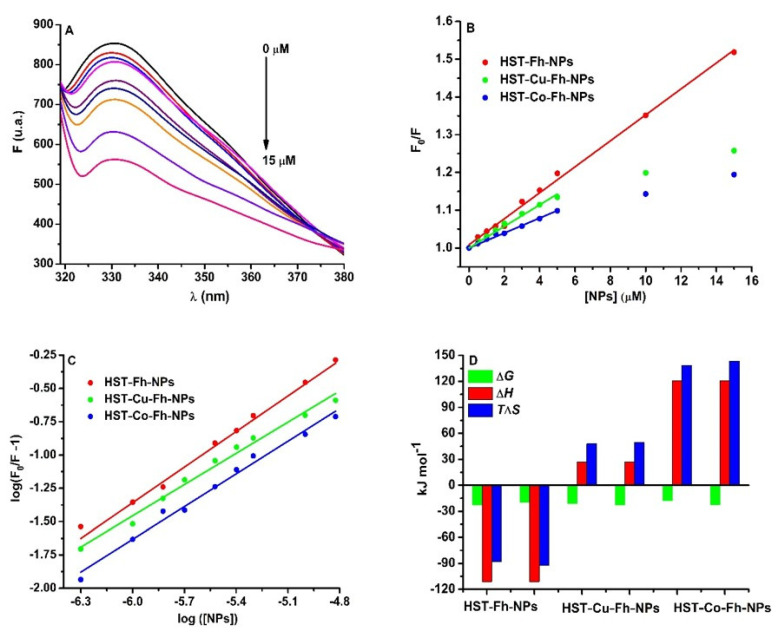
(**A**): The fluorescence emission spectra of HST (3 µM) in the presence of Fh-NPs (0–15 µM), (**B**) the Stern-Volmer representation of F_0_/F vs. [NPs] for HST in complex with Fh-NPs (●red), Cu-Fh-NPs (●green), and Co-Fh-NPs (●blue) (0–15 µM), and (**C**): the Scatchard plot of log (F_0_/F^−1^) against log[NPs]. All samples were prepared in 100 mM HEPES buffer, at pH 7.35 and were recorded at 25 °C. (**D**) The thermodynamic parameters for the binding of the Fh-NPs, Cu-Fh-NPs, and Co-Fh-NPs with HST.

**Figure 2 ijms-22-07034-f002:**
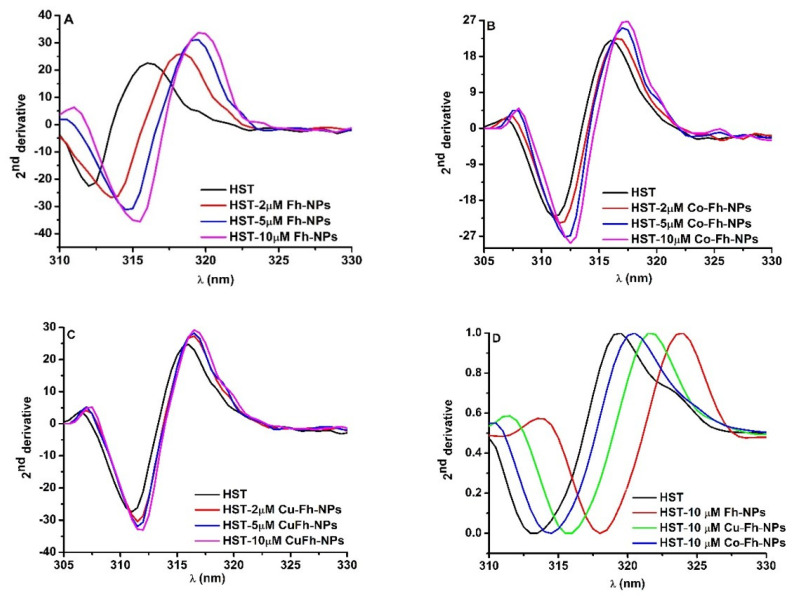
(**A**–**C**): The second derivatives of the Trp emission spectra of HST at different Fh-NPs, Cu-Fh-NPs and Co-Fh-NPs concentrations (0, 2, 5 and 10 μM) and (**D**): The second derivative of the Trp emission spectra of HST (black curve) and HST in the presence of 10 μM Fh-NPs (red curve), Cu-Fh-NPs (green curve) and Co-Fh-NPs (blue curve).

**Figure 3 ijms-22-07034-f003:**
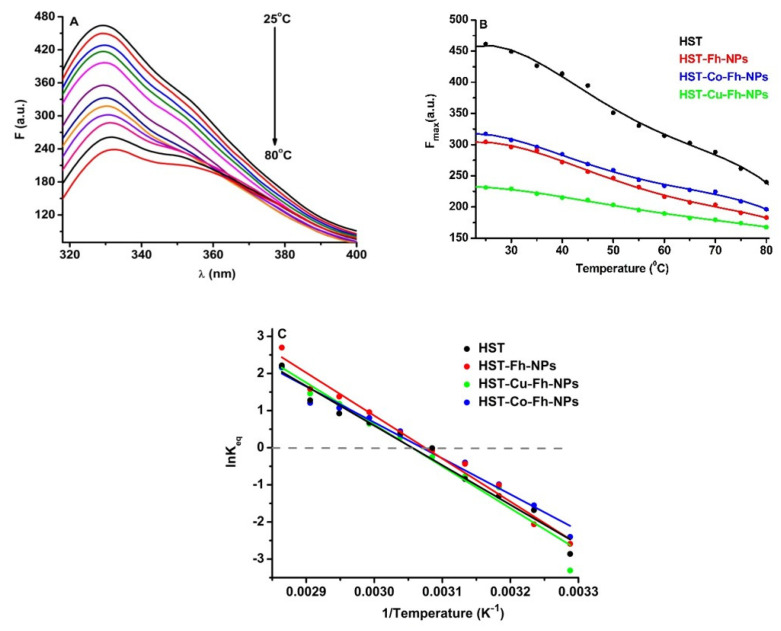
(**A**): The fluorescence corresponding to the thermal denaturation processes of HST (3 µM), (**B**): Maxima of the fluorescence of HST in the absence and presence of simple and doped Fh-NPs (10 µM), and (**C**): Graphical representation of ln*K_eq_* vs. 1/*T* for the denaturation of HST structure the absence and presence of simple and doped Fh-NPs.

**Figure 4 ijms-22-07034-f004:**
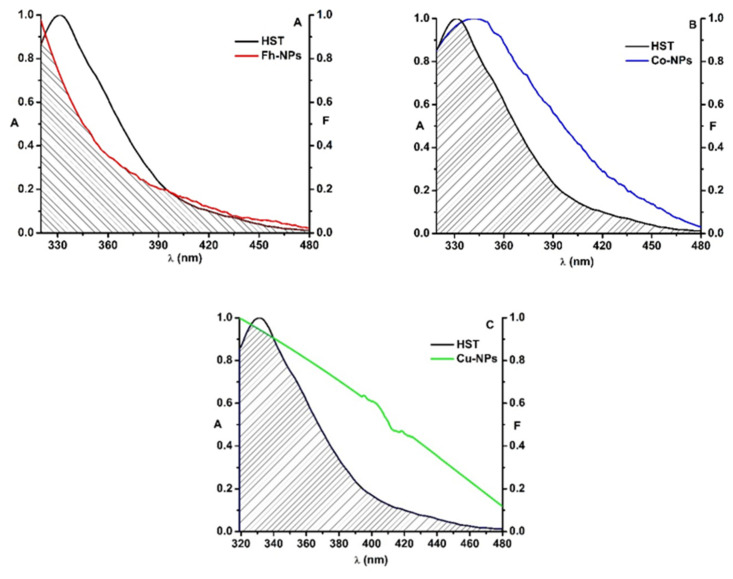
The overlap of the fluorescence emission spectrum of HST (black) and the absorption spectrum of Fh-NPs (red, **A**), Co-Fh-NPs (blue, **B**), Cu-Fh-NPs (green, **C**) and ([HST]: [NPs] = 1:1).

**Figure 5 ijms-22-07034-f005:**
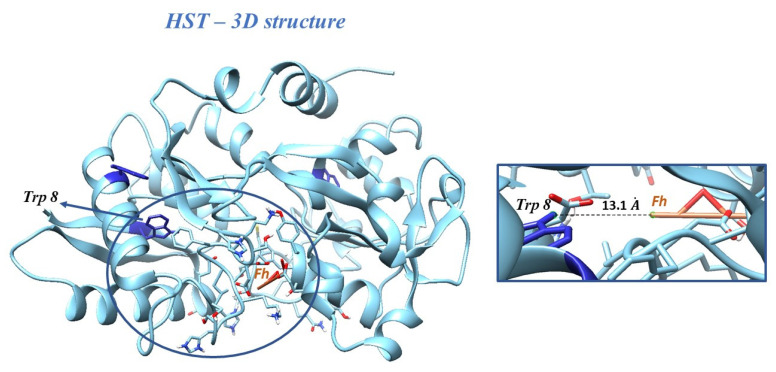
Docking of Fh into the structure of HST. Inset: the binding site of HST for Fh, with the Trp 8 residue and the distance between Trp 8 and Fh.

**Table 1 ijms-22-07034-t001:** The binding parameters of the interaction of Fh-NPs with HST.

*T* (°C)	*K_SV_* × 10^4^ (M^−1^)	*K_q_* × 10^13^ (M^−1^ s^−1^)	*K_b_* × 10^3^ (M^−1^)	Δ*G* (kJ mol^−1^)	Δ*H* (kJ mol^−1^)	*T*Δ*S* (kJ mol^−1^)
**HST-Fh**						
25	3.44 ± 0.05	1.03 ± 0.01	9.54 ± 0.08	−22.71 ± 0.28	−111.02 ± 0.29	−88.31 ± 0.31
35	3.43 ± 0.06	1.02 ± 0.02	2.23 ± 0.02	−19.75 ± 0.38		−92.27 ± 0.29
**HST-Cu-Fh**						
25	3.03 ± 0.04	0.90 ± 0.02	5.10 ± 0.02	−21.16 ± 0.21	26.76 ± 0.32	47.92 ± 0.29
35	3.85 ± 0.09	1.15 ± 0.03	7.24 ± 0.04	−22.76 ± 0.36		49.52 ± 0.34
**HST-Co-Fh**						
25	2.45 ± 0.21	0.73 ± 0.02	1.30 ± 0.16	−17.77 ± 0.31	120.91 ± 0.37	138.6 ± 0.39
35	2.97 ± 0.19	0.88 ± 0.02	6.33 ± 0.25	−22.54 ± 0.37		143.4 ± 0.36

**Table 2 ijms-22-07034-t002:** Thermodynamic fingerprint of HST denaturation in the absence and presence of simple and doped Fh-NPs.

Sample	ΔH (kJ mol^−1^)	ΔS (J mol^−1^ K^−1^)	T_m_ (°C)
HST	88.618 ± 0.35	270.79 ± 0.26	54.10 ± 0.88
HST-Fh-NPs	96.209 ± 0.15	295.80 ± 0.18	52.10 ± 1.09
HST-Co-Fh-NPs	80.498 ± 0.22	247.09 ± 0.24	52.63 ± 1.61
HST-Cu-Fh-NPs	93.786 ± 0.23	286.55 ± 0.15	54.14 ± 0.97

**Table 3 ijms-22-07034-t003:** FRET parameters for the interaction of simple and doped Fh-NPs with HST.

Sample	J × 10^14^ (M^−1^ cm^−1^ nm^4^)	E	R_0_ (nm)	r (nm)
Fh-NPs-HST	3.83	0.12	2.29	3.18
Cu-Fh-NPs-HST	2.09	0.08	2.07	3.10
Co-Fh-NPs-HST	1.82	0.05	2.02	3.23

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
