# Peer review of "Interactions of Chemically Synthesized Ferrihydrite Nanoparticles with Human Serum Transferrin: Insights from Fluorescence Spectroscopic Studies"

_ijms, 2021, doi:10.3390/ijms22137034_

Round 1
Reviewer 1 Report
The manuscript entitled " Interactions of Chemically Synthesized Ferrihydrite Nanoparticles with Human Serum Transferrin: Insights from Fluorescence Spectroscopic Studies" is interesting and the results were reasonable. However, please consider my comments and suggestions: before a further consideration.
- It is suggested the authors to rearrange the introduction section for a better flow and cohesiveness as a background to the readers.
- Three different types of nanoparticles, Fh-NPs, 16
Cu-Fh-NPs, and Co-Fh-NPs, have been used in this study however, no information about particle size and surface characterization have shown in this work. Authors cited their previous work for particle size characterization by means of atomic force microscopy (AFM), transmission electron microscopy (TEM), scanning electron microscopy (SEM), and small-angle neutron scattering methods (SANS). However, i suggest that show the surface characterization data and particle size characterization data in the manuscript or supporting information part. - It was reported that Fh-NPs interacted with HST with low affinity, and the binding is driven by hydrogen bonding and van der Waals forces for simple Fh-NPs and by hydrophobic interactions for Cu-Fh-NPs and Co-Fh-NPs binding, respectively. However, it is not clearly mentioned how authors made this conclusion. Please include more detailed information.
- There are a few typos through out the manuscript and please correct.
Author Response
Dear Sir,
Please find enclosed the revision of our manuscript ijms-1261687 entitled “Interactions of chemically synthesized ferrihydrite nanoparticles with human serum transferrin: Insights from fluorescence spectroscopic studies”.
Thank you for your evaluation and comments on the manuscript. We have revised the manuscript to address all the concerns and we have added Supplementary data, and extensively re-written the manuscript.
We sincerely hope that you will now find this manuscript acceptable for publication in the International Journal of Molecular Science.
Yours,
Claudia Chilom

Reviewer 2 Report
The manuscript of Claudia Chilom et al details the interactions of ferrihydrite NPs and Human Serum Transferrin. After reading it I have the impression that the results are leftovers from previously published studies. However, the manuscript may contain enough novelty to be published in IJMS, but only after a major revision. The manuscript must be significantly rewritten, since it contains a number of errors even conclusional ones.
My comments to the authors are as follows:
Major issues:
The Introduction is too short and needs to be extended, for example with the application of similar functionalized nanoparticles in cancer therapy and especially the quenching phenomenon of proteins in the presence of different quenchers.
The authors state that there is a positive deviation in the case of the two doped NPs. This divergence is called downward curvature and a very good explanation of it can be found in https://www.researchgate.net/publication/7287837_Steady-state_fluorescence_quenching_applications_for_studying_protein_structure_and_dynamics
http://dx.doi.org/10.1016/j.jphotobiol.2005.12.017
The downward curvature is characteristic for two fluorophore populations, one of which is not accessible for the quencher, that is why the remaining fluorescence is higher than that of expected from the concentration of the quencher.
Nagy et al present an excellent example in their article where the downward curvature of the Stern-Volmer curve was explained by an H-bonded nonfluorescent and a competing pi-stacked fluorescent complex see Fig 9. In https://doi.org/10.1002/cphc.201402310 (Solvatochromic Study of Highly Fluorescent Alkylated Isocyanonaphthalenes, Their π-Stacking, Hydrogen-Bonding Complexation, and Quenching with Pyridine)
These two papers should be referenced in the Introduction or in the Results sections.
Based on the two above cited articles the quenching cannot be both static and dynamic, because it would result in an upward curvature, since fluorescence would decrease quicker with the increased quencher concentration owing to the presence of a dark complex. This issue needs to be addressed in the whole manuscript.
Since the downward curvature is present only in the case of the doped NPs, in my opinion the real explanation may be the complexation of HST with the Cu or Co ions leaking from the NPs. These complexes would result in conformational changes where the NPs could no longer quench the fluorescence, because the tryptophans or tyrosines would be locked in an inaccessible position.
Other/minor issues:
Line 38:
“HST is in a concentration of 30% in the blood but it can be affected by various types of diseases [Sepand et al., 2020].”
This sentence needs to be rephrased. 30% is too high concentration in the blood.
Line 64: In the formula of Fe(NO3)3·x 9H2O, there are a dot and an x indicating crystalline water. One of them should be deleted.
Line 65 and 69: There is no such thing as neutral pH. There is either pH=7(in water) or neutral medium.
Line 84: 3/10 μM is this 0.3 μM? If yes please exchange.
In equation 4 (also valid for the other eqs.) please use the equation editor of word and do not insert the eq as a picture. This is also valid for the variables in lines 129-132.
In figure 1 please use different markers for the different nanoparticles, because in B&W prints they cannot be differentiated. In the figure caption there is a black dot for all three NPs.
In Table 1, How did You calculate the Kb and the thermodynamic (ΔG, ΔH, ΔS) values? What is the error of these measurements? Did You use only two temperatures for calculation?
Line 175, please explain in more detail what is the basis of the following statements: “in the case of Fh-NPs the main forces of the binding are hydrogen bonding and van der Waals forces (ΔH < 0 and ΔS < 0). For the binding of Cu-Fh-NPs and Co-Fh-NPs to HST (Table 1), processes are mainly driven by hydrophobic interactions(ΔH> 0 and ΔS> 0).“
In Table 2, The unit of ΔH is only kJ mol-1 please correct! Moreover, the precision of the values in Table 2 seem too high. What was the error of the measurements?
The main characterization results of the NPs should be included in the experimental section. We have no information on the size, shape and polydispersity index of the NPs.
In Table 3 why do You say that “As the rule 0.5R0< r< 1.5R0 does not hold in the case of the interaction of doped Fh-NPs”? 2,07*1,5=3,105 it is within r< 1.5R0 and the other value is also very close to 1.5R0.
Author Response

(The authors gave the same response as above.)

Reviewer 3 Report
The research article under title “Interactions of Chemically Synthesized Ferrihydrite Nanoparticles with Human Serum Transferrin: Insights from Fluorescence Spectroscopic Studies” investigates the interactions of ferrihydrite nanoparticles with HSF by spectrofluorometry and molecular docking. Although there is plenty of results presented and discussed, there are some additional points that should be addressed by the authors. Therefore my recommendation is a MAJOR REVISION.
- Human serum transferrin (HST) is a glycoprotein involved in iron transport that may be a candidate for functionalizing nanoparticles - Human serum transferrin (HST) is a glycoprotein involved in iron transport that may be a candidate for FUNCTIONALIZED nanoparticles
- HST is in a concentration of 30 % in the blood – what is this concentration? One would not expect to have HST in such a high concentration in blood
- What is the structure of Fh when it is written in line 45 that it is iron oxide?
- The authors should explain in the materials section what was the role of Cu and Co in the structure of Fh. I believe that this is an important thing for the explanation of the mechanism of binding and thermodynamic parameters.
- Some of the main structural characteristics explained by these experimental techniques should also be included in the materials section.
- In the paper several times it is said that the data is not shown. I would suggest that the authors present the data in the Supplementary material as this is important for understanding the binding mechanism.
- How was Kq calculated?
- In the manuscript, the authors do not give any possible explanation of the effect of Co and Cu. For example, in lines 158-165 the binding affinity is just stated. The effect of Co and Cu would significantly increase the quality of the work and doping would make sense.
- Figure 1D should be removed as it is not very informative and it presents the same data as Table 1
- I would also suggest removing paragraphs between lines 174 and 178 as this is an oversimplified representation of the different interactions. Or at least the authors should give some references and also explain why obtained nanoparticles would interact with surrounding amino acids by the hydrophobic interactions, especially if doped with Co and Cu.
- The second derivative spectra are given and the affinity was calculated between lines 187 and 192 but still, the possible explanation is missing. The authors just present the data and results, but it would be beneficial to have at least some possible explanation that is in line with the results.
- Line 214 states that The obtained values of Tm suggest that simple and doped Fh-NPs to HST cause small alterations in the protein structure – this is too general for the paper and authors should elaborate more on this. The differences in thermodynamic parameters are negligible and they can be contributed to the experimental error and the influence of the added ions is not visible.
- The results of the FRET mechanism should also be connected with the structural changes after doping.
- How was the binding site of HST determined? Just stating the residues does not bear any significant information. The authors should connect with some of the previously published results. The value of the binding energy in line 264 should be given in the same units that are used throughout the paper.
- The chosen structure of Ff should be clearly given as it is not clear from the manuscript and figures which structure was used.
Author Response

(The authors gave the same response as above.)

Round 2
Reviewer 1 Report
I believe the manuscript has been sufficiently improved to warrant publication in IJMS.
Author Response
Thank you for carefully reviewing the manuscript.
With consideration,
Claudia Chilom
Reviewer 2 Report
Dear Authors!
Thank You for the corrections! However, a few errors still remain in the text:
In the Abstract: s "for Cu-Fh-NdockingPs" a space is missing
Introduction: "The doping nanoparticles" I think an "of" is missing
Experimental: "Structural characterization of ferrihydrite samples by Transmission Electron Microscopy (TEM) showed that once doped with Co and Cu ferrihydrite particles increase average size from ~2.5 nm in the case of Fh-NPs to ~3.5 nm in the case of doped NPs [Stolyar et al., 2017]. " The meaning of this sentence is unclear, please rephrase! In the previous sentence completely different sizes are listed.
Page 5 bottom: "the Stern-Volmer plot shows downward curvatures of the Stern-Volmerplot [Nagy et al., 2014]." This part of the sentence should read as: "the Stern-Volmer plots show downward curvatures [Nagy et al., 2014]."
In Table 1. the unit of TdeltaS is only "kJ*mol-1"!
Author Response
Thank you for carefully reviewing the manuscript. We hope that, as a result of the corrections, its quality has improved.
Regards,
Claudia Chilom

Reviewer 3 Report
the authors have answered to all of the remarks by the reviewer.
Author Response

(The authors gave the same response as above.)
